# FAST MULTI-MODE ADAPTIVE GENERATIVE DISTILLATION FOR CONTINUALLY LEARNING DIFFUSION MODELS

## ABSTRACT

Diffusion models are powerful generative models, but their computational demands, vulnerability to catastrophic forgetting, and class imbalance in generated data pose significant challenges in continual learning scenarios. In this paper, we introduce Fast Multi-Mode Adaptive Generative Distillation (MAGD), a novel approach designed to address these three core challenges. MAGD combines generative replay and knowledge distillation, enhancing the continual training of diffusion models through three key innovations: (1) Noisy Intermediate Generative Distillation (NIGD), which leverages intermediate noisy images during the reverse diffusion process to improve data utility and preserve image quality without additional computational costs; (2) Class-guided generative distillation (CGGD), which uses classifier guidance to ensure balanced class representation in generated images, addressing the issue of class imbalance in traditional methods; and (3) Signal-Guided Generative Distillation (SGGD), which reduces computational overhead while maintaining image clarity through the reuse of the model's denoising capabilities across tasks. Our experimental results on Fashion-MNIST, CIFAR-10, and CIFAR-100 demonstrate that MAGD significantly outperforms existing methods in both image quality, measured by Fréchet Inception Distance (FID), and class balance, measured by Kullback-Leibler Divergence (KLD). Moreover, MAGD achieves competitive results with far fewer generation steps compared to traditional methods, making it a practical solution for real-life continual learning applications.

## 1 INTRODUCTION

Diffusion models have become a cornerstone in the field of generative modeling due to their exceptional ability to produce high-quality images and achieve state-of-the-art performance across various benchmarks Ho et al. (2020); Dhariwal & Nichol (2021). Despite their success, training diffusion models presents significant challenges. Chief among these is their **computational intensity**, as generating data typically requires simulating thousands of denoising steps Ho et al. (2020), making them impractical for applications requiring rapid updates or operation under limited computational resources. While some methods Song et al. (2020); Salimans & Ho (2022); Song et al. (2023) have proposed strategies to reduce computational costs by decreasing the number of generation steps, these approaches are designed primarily for offline scenarios and fail to address settings where data distributions shift over time, requiring models to be periodically updated—a scenario known as continual learning.

Current works Masip et al. (2023); Gao & Liu (2023); Meng et al. (2024); Jodelet et al. (2023) have applied diffusion models for *generative replay* in class-incremental learning scenarios and have primarily focused on improving classification accuracy by using diffusion models to replace past datasets. However, they do not adequately address the challenges of efficiently training the diffusion model itself in a continual learning framework, where catastrophic forgetting can significantly impact the model's generative capabilities. Masip et al. (2023) introduces the notion of *generative distillation* which improves over *generative replay* by distilling the noise predictions of the teacher model rather than relying on its synthesized images but it overlooks class balance and the rich knowledge carried by the reverse diffusion process used in the teacher model for data generation.

Thus, effectively training diffusion models in continual learning contexts is both an important and challenging problem. The first challenge in this context is the **degradation of image quality** over time. As the model updates with new data, it tends to forget how to generate earlier data distributions, resulting in deteriorated, blurry, or unusable images, as illustrated in Fig. 1 Lesort et al. (2018); Masip et al. (2023); Meng et al. (2024). Additionally, the **computational demands** of the image generation process for generative replay make it unsuitable for applications requiring rapid adaptation to new data Ho et al. (2020); Song et al. (2020). Finally, there is a significant issue of **class imbalance** in generated data, as shown in Fig. 2. Diffusion models often fail to generate an equitable representation of all classes, negatively affecting the performance of downstream applications due to biased training samples.

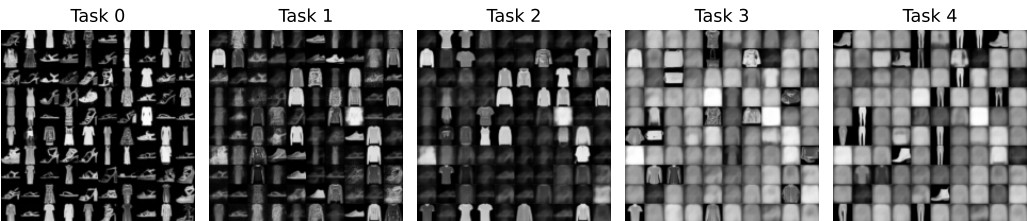

Figure 1: Images generated using generative replay Shin et al. (2017) after training each task in the split Fashion-MNIST scenario.

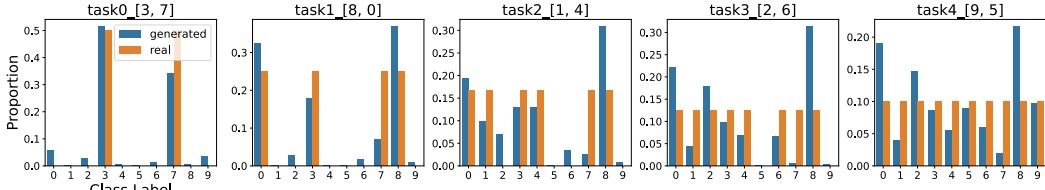

Figure 2: Comparison between images generated by an unconditional diffusion model and real training data in a class-incremental learning scenario with 5 tasks on the Fashion-MNIST dataset. The histogram shows an imbalanced distribution, especially for class 7. In task 0, the model learns to generate images for classes 3 and 7, but the proportion of generated images for class 7 is already low. This imbalance worsens with more tasks, eventually resulting in almost no images generated for class 7.

In this work, we introduce **Fast Multi-Mode Adaptive Generative Distillation (MAGD)**, a novel framework designed to tackle the key challenges in continually training diffusion models. MAGD seamlessly integrates the strengths of *generative replay* and *knowledge distillation* to efficiently and comprehensively transfer learned knowledge across tasks, addressing critical issues such as image quality degradation, computational inefficiency, and class imbalance. Our approach builds on three key innovations, all centered around the concept of *generative distillation*:

- **Noisy Intermediate Generative Distillation** (NIGD): We propose an enhanced distillation strategy that utilizes intermediate noisy images ($\hat{x}\tau$) directly, bypassing the need to generate clean images ($x_0$) before reapplying noise. This method leverages the entire sequence of noisy images produced during the reverse diffusion process, allowing knowledge to be distilled at every stage of the generation process. This approach maximizes data utility without adding computational overhead, preserving image quality over successive tasks and improving overall training efficiency.

- **Class-Guided Generative Distillation** (CGGD): Class imbalance in generated data is a significant challenge in continual learning with diffusion models. To address this, we incorporate classifier guidance into the distillation process, ensuring that the diffusion model

generates a balanced representation of each class. This class-guided distillation prevents the model from over-representing certain classes, a common issue in traditional generative replay methods, and supports more robust performance in downstream classification tasks.

- **Signal-Guided Generative Distillation (SGGD)**: Inspired by recent studies on the denoising capabilities of diffusion models Deja et al. (2022); Zajac et al. (2023), we separate the denoising and generative stages, allowing for efficient reuse of the denoising component across tasks. This signal-guided approach reduces computational costs while maintaining image clarity, enabling the model to adapt to new tasks without sacrificing the quality of generated images—an issue typically seen in continual learning frameworks.

We evaluate our methods on three widely-used datasets in the continual learning community—Fashion-MNIST, CIFAR-10, and CIFAR-100. Our experimental results demonstrate that our comprehensive generative distillation framework significantly outperforms traditional methods that replace past datasets with diffusion models. Specifically, our approach yields improvements in both Fréchet Inception Distance (FID), indicating effective preservation of image quality over time, and Kullback-Leibler Divergence (KLD), demonstrating better class distribution in generated images.

Notably, our method achieves competitive performance with only a few generation steps (5 steps for Fashion-MNIST and 20 steps for CIFAR-10 and CIFAR-100), compared to the 1,000 steps required by methods like DDGR Gao & Liu (2023). This substantial reduction in computational requirements makes our approach suitable for real-life continual learning applications.

In summary, our contributions are as follows:

- **Mitigating Image Quality Degradation:** We mitigate image quality degradation over time through *Noisy Intermediate Generative Distillation (NIGD)*. By utilizing generated noisy images at intermediate steps for distillation and leveraging the powerful denoising capabilities of diffusion models, we ensure high-quality image generation across multiple tasks.

- **Addressing Class Imbalance:** With *Class-Guided Generative Distillation (CGGD)*, we incorporate classifier guidance into the generative distillation process. This ensures balanced image generation across classes, preventing class dominance or bias and improving the performance of downstream classifiers in continual learning scenarios.

- **Efficient Continual Training of Diffusion Models:** Our framework, *Signal-Guided Generative Distillation (SGGD)*, introduces a method for efficiently training diffusion models by reusing the denoising components across tasks. This significantly reduces computational costs while maintaining the ability to generate high-quality images throughout continual learning tasks.

## 2 RELATED WORK

### 2.1 CONTINUAL LEARNING

Continual Learning has emerged as a significant challenge, focusing on enabling models to learn new knowledge over time without forgetting previously acquired knowledge. To address the issue of catastrophic forgetting, many recent approaches Rebuffi et al. (2016); Wu et al. (2019); Douillard et al. (2020); Wang et al. (2023) involve storing training data from earlier classes as exemplars and replaying them while learning new tasks. While exemplars are beneficial for reinforcing past knowledge, their use may be impractical due to privacy concerns, legal restrictions, and limited memory resources on devices.

To overcome these limitations, some researchers have proposed using generative models Shin et al. (2017); Lesort et al. (2018); Zhai et al. (2019); Wu et al. (2018) to synthesize data from previous classes instead of storing real data. These methods typically employ Generative Adversarial Networks (GANs) Goodfellow et al. (2014) or Variational Autoencoders (VAEs) Kingma & Welling (2013) as image generators. However, in the context of continual learning, it is crucial to continually update the generative model itself. When trained solely on its own generated data due to a lack of real data from earlier tasks, the quality of the generated images tends to progressively deteriorate, often resulting in blurry outputs, as shown in Fig. 1.

In this paper, we propose a novel approach that distills knowledge from all generated noisy images during the diffusion model's generation process. This method helps to reduce forgetting by leveraging the inherent properties of diffusion models, maintaining the quality of generated data, and addressing the computational challenges associated with continual learning scenarios.

## 2.2 DIFFUSION MODELS IN CONTINUAL LEARNING

Diffusion models Ho et al. (2020) have gained significant attention for their performance on various benchmarks Dhariwal & Nichol (2021), though their high computational cost remains a challenge. To address this, methods like DDIM Song et al. (2020), progressive distillation Salimans & Ho (2022), and consistency models Song et al. (2023) have been proposed. In this paper, we adopt DDIM for generation, as it deterministically maps noise to original data while preserving noise distribution, making it ideal for continual learning.

Although recent works have applied diffusion models in continual learning Zajac et al. (2023); Masip et al. (2023); Gao & Liu (2023), they mostly use the diffusion model to replace the replay buffer in memory-based methods, employing simple continual training strategies. Jodelet et al. (2023) uses a pretrained Stable Diffusion model as a fixed supplementary replay buffer throughout training. Masip et al. (2023) introduces the concept of *generative distillation*, which enhances *generative replay* by distilling the noise predictions of the teacher model instead of relying on its synthesized images. However, it overlooks both the issue of class balance and the rich knowledge embedded in the reverse diffusion process used by the teacher model for data generation. Meng et al. (2024) achieves the best results but requires a separate diffusion model for each task, making it inefficient. These methods fail to explore continual training by leveraging the diffusion model's inherent properties.

The high computational cost of using diffusion models for image generation is a major concern in real-life applications. To address this, we propose a novel approach that distills knowledge from both generated and training images, Gaussian noise, and all intermediate noisy images produced during generation. This comprehensive distillation strategy leverages diffusion model properties for more efficient training, lowering computational costs and mitigating catastrophic forgetting and image quality degradation.

## 2.3 KNOWLEDGE DISTILLATION OF DIFFUSION MODELS

The primary challenge in using diffusion models for real-world applications lies in the high computational cost of generating images, which typically requires thousands of steps to denoise the initial noise. To reduce these costs, some studies, such as Salimans & Ho (2022); Song et al. (2023); Zheng et al. (2022), have focused on distilling knowledge from pretrained diffusion models to develop supplementary models that denoise with significantly fewer steps. However, these methods face limitations in continual learning scenarios for two main reasons: 1) they rely on access to data from previous tasks, which is not available in our case; and 2) they assume the pretrained model remains static, whereas in our scenarios, the model must evolve to incorporate new information as it becomes available. Thus, simply distilling a pretrained model into a more efficient version is inadequate.

## 3 PROBLEM FORMULATION

In our paper, we consider the setting of class incremental learning as mentioned in van de Ven & Tolias (2019), consisting of $N$ tasks. The dataset is denoted as $\mathbb{D} = \{\mathbb{D}_k\}_{k=0}^{k=N-1}$, where, $\mathbb{D}_k = \{\boldsymbol{X}_k, \boldsymbol{Y}_k, \boldsymbol{C}_k\}$ contains the dataset used in the task $k$. Here, $\boldsymbol{X}_k$ represents the training images, $\boldsymbol{Y}_k$ represents the class labels, $\boldsymbol{C}_k$ contains the unique class labels in task $k$, and $d_k$ represents the data length In the class-incremental learning scenario, $\boldsymbol{C}_i \bigcap \boldsymbol{C}_j = \emptyset$. The diffusion model is denoted as $\theta$ with $T$ generation steps. $\boldsymbol{M}$ represent the memory set which store the true images or generated images.Then the global objective from task 0 to current $n$ can be denoted as:

$$L^* = \sum_{k=0}^{n} l_k \quad ; \quad l_k = \frac{1}{d_k T} \sum_{\boldsymbol{x}_0 \in \mathbb{D}_k} \sum_{t=1}^{T} ||\epsilon_t - \theta(\boldsymbol{x}_t, t)||^2 \tag{1}$$

However, within the setting of class-incremental learning, the model cannot access to all data from previous tasks. Thus, the objective at task $n$ can be formulated as:

$$L_n = l_n + l_M \quad ; \quad l_M = \frac{1}{d_M T} \sum_{\boldsymbol{x}_0 \in \boldsymbol{M}} \sum_{t=1}^{T} ||\epsilon_t - \theta(\boldsymbol{x}_t, t)||^2 \tag{2}$$

## 4 METHODOLOGY

### 4.1 GENERATIVE REPLAY AND GENERATIVE DISTILLATION

In this section, we introduce the mechanisms of Generative Replay (DGR) and its variant, Generative Distillation(DGR-distill), as outlined in Algorithm 1. Both methods serve as baselines for our discussion. During training each batch, it involves creating a memory batch, denoted as $\mathbf{X}_r$, and adding noise $\epsilon_r$ corresponding to step $t_r$. The primary distinction between DGR and DGR-distill lies in the computation of the replay loss $l_r$. In DGR, the model is trained to predict the noise $\epsilon_r$. In contrast, DGR-distill trains the model to approximate the previous output $\theta^{k-1}(\mathbf{X}_r, t_r)$.

In the following, starting from DGR-distill, we introduce the proposed comprehensive generative distillations in three significant ways to enhance its performance and applicability: 1) We modify the generation process of $\mathbf{X}_r$ with **NIGD** and **CGGD**. 2) We revise the calculation of the replay loss $l_r$ with **NIGD**. 3) We refine how we sample $t_r$ with **SGGD**.

### 4.2 NOISY INTERMEDIATE GENERATIVE DISTILLATION (**NIGD**)

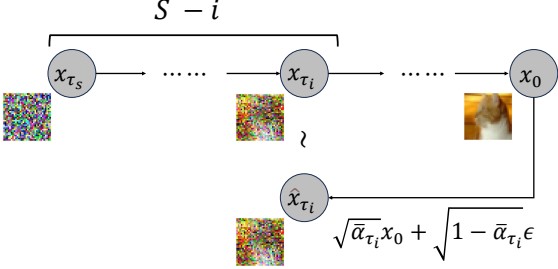

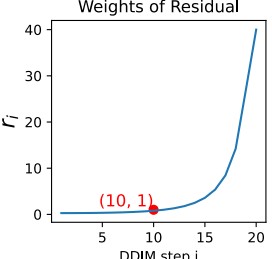

Figure 3: Comparison of adding noise to original images, denoted as $\hat{\boldsymbol{x}}_{\tau_i}$, versus directly generating noisy images, denoted as $\boldsymbol{x}_{\tau_i}$

Figure 4: Evaluation of $\boldsymbol{r}_i$ of 20 generation steps

To rapidly generate images, we use a DDIM schedulerSong et al. (2020). Rather than employing all $T$ steps, this method utilizes a subset $\boldsymbol{x}_{\tau_1}, \boldsymbol{x}_{\tau_2}, \ldots, \boldsymbol{x}_{\tau_s}$, where $\tau$ represents an increasing subsequence of $[1, \ldots, T]$ with length $S$. Assuming the trained diffusion model is denoted by $\theta$, we then proceed with the reverse process by :

$$\boldsymbol{x}_{\tau_i} = \sqrt{\bar{\alpha}_{\tau_i}} * \frac{\boldsymbol{x}_{\tau_{i+1}} - \sqrt{1 - \bar{\alpha}_{\tau_{i+1}}} \theta(\boldsymbol{x}_{\tau_{i+1}})}{\sqrt{\bar{\alpha}_{\tau_{i+1}}}} + \sqrt{1 - \bar{\alpha}_{\tau_i}} \theta(\boldsymbol{x}_{\tau_{i+1}}) \tag{3}$$

After completing $S$ steps of the reverse process, we obtain the generated image denoted as $\boldsymbol{x}_0$. In the forward process, the distribution $q(\boldsymbol{x}_{\tau_i}|\boldsymbol{x}_0) = \mathcal{N}(\boldsymbol{x}_{\tau_i}; \sqrt{\bar{\alpha}_{\tau_i}}\boldsymbol{x}_0, (1 - \bar{\alpha}\tau_i)\boldsymbol{I})$ describes how the image $\boldsymbol{x}_0$ transitions to its noisy versions. Specifically, we derive the noisy images $\hat{\boldsymbol{x}}_{\tau_i}$ directly from $\boldsymbol{x}_0$.

$$\hat{\boldsymbol{x}}_{\tau_i} = \sqrt{\bar{\alpha_{\tau_i}}}\boldsymbol{x}_0 + \sqrt{(1 - \bar{\alpha}_{\tau_i})}\epsilon \tag{4}$$

We then derive (full demonstration is detailed in Appendix A.3) :

$$\hat{x}_{\tau_i} - x_{\tau_i} = \sum_{j=i}^{1}(r_j\theta(x_{\tau_j})) \quad ; \quad r_j = \sqrt{\bar{\alpha}_{\tau_i}}(\sqrt{\frac{1-\bar{\alpha}_{\tau_{j-1}}}{\bar{\alpha}_{\tau_{j-1}}}} - \sqrt{\frac{1-\bar{\alpha}_{\tau_j}}{\bar{\alpha}_{\tau_j}}}) \tag{5}$$

From Eq. (5), we observe that the difference between the noisy image $\hat{x}_\tau$, derived from the generated $x_0$ by adding noise, and the directly generated noisy image $x_\tau$ depends solely on the generation steps from $\tau_i$ to $\tau_1$. As shown in Fig. 4, we evaluate the values of $r_j$ when $\tau_i = 500$ ($i = 10$) for 20 steps of DDIM. We find that for all $j < 10$, the residuals $r_j$ are smaller than 1 and significantly lower than $r_j$ for $j > 10$. This indicates that the residual component is relatively weaker compared to the noisy image $x_{\tau_i}$.

In our continual learning scenario, we utilize the previously trained diffusion model on task $k - 1$, denoted as $\theta^{k-1}$, as our teacher model. Our objective is to train the new model $\theta^k$ by distilling knowledge from the teacher. Therefore, for any given $\tau_i$, we require:

$$\theta^k(x_{\tau_{i-1}}|x_{\tau_i}, \tau_i) = \theta^{k-1}(x_{\tau_{i-1}}|x_{\tau_i}, \tau_i) \tag{6}$$

However, during the generation process, the previously trained model only generates $x_{\tau_i}$ without access to $\hat{x}_{\tau_i}$. Therefore, $x_{\tau_i}$ is sufficient for distillation from the previous model.

We demonstrate that in a $S$-step DDIM generation process, for any given $\tau_i$, the directly generated noisy image $x_{\tau_i}$ and the noisy image $\hat{x}_{\tau_i}$, obtained by adding noise to $x_0$, are equivalent for distillation purposes, as shown in Fig. 3.

This study suggests an equivalence between two methods of obtaining $\mathbf{x}_t$:

1. **Two-Stage Approach**: Generate $x_0$ using $S$ generation steps, and then add noise corresponding to the time step $t$ to obtain the noisy image $x_t$.

2. **Direct Approach**: Directly generate $x_t$ by using $S = \frac{\tau_s - t}{\Delta\tau}$ generation steps.

In practice, we use $S$ generation steps and distill from all intermediate noisy images generated throughout the inverse process. This approach suggests that we can efficiently enhance the quality of the generated noisy images without increasing computational costs.

### 4.3 CLASS-GUIDED GENERATIVE DISTILLATION(**CGGD**)

In the continual learning community, several studies Wu et al. (2019); Lin et al. (2023a); Zhao et al. (2019) have explored catastrophic forgetting in classification problems. These studies found that the weights corresponding to previously learned classes tend to decrease, while the weights associated with the current class increase, as the current class dominates the training data. As a result, the model tends to overpredict the current class at the expense of the previous ones.

In diffusion models, there are primarily two types of conditioned models: classifier-guided Dhariwal & Nichol (2021) and classifier-free Ho (2022). The classifier-free model typically requires higher computational costs during both training and inference. In this paper, we opt for a classifier-guided diffusion model, where a supplementary classifier is trained to guide the inference process. Our method generates an equal number of samples for each learned class, helping to balance the generated images for effective distillation. For training task k, we employ the following process to generate a noisy image for distillation from the model trained on the previous task, denoted as $\theta^{k-1}$ for the diffusion model and $g^{k-1}$ for the classifier. First, we randomly select a class label $y$ from a uniform distribution. Then one-step update process is defined as follows using cross-entropy loss CE:

$$\hat{\epsilon} = \theta^{k-1}(x_{\tau_{i+1}}, \tau_{i+1}) - \sqrt{1-\bar{\alpha}_{\tau_{i+1}}}\nabla_{x_{\tau_{i+1}}}CE(g^{k-1}(x_{\tau_{i+1}}, \tau_{i+1}), y)$$

$$x_{\tau_i} = \sqrt{\bar{\alpha}_{\tau_i}} * \frac{x_{\tau_{i+1}} - \sqrt{1-\bar{\alpha}_{\tau_{i+1}}}\hat{\epsilon}}{\sqrt{\bar{\alpha}_{\tau_{i+1}}}} + \sqrt{1-\bar{\alpha}_{\tau_i}}\hat{\epsilon} \tag{7}$$

## 4.4 SIGNAL-GUIDED GENERATIVE DISTILLATION (**SGGD**)

Deja et al. (2022) discovered that a diffusion model operates in two distinct phases based on the time steps ($t$): as a denoiser for refining corrupted images into final samples when $t$ is small, and as a generator for creating images from noise when $t$ is larger. Their research shows robust generalization across datasets such as CIFAR-10 and CelebA, particularly in the early stages of diffusion (when $(t/T < 0.1)$), as illustrated in Fig. 3 of Deja et al. (2022)

The use of solely generated images for training in continual learning scenarios, as discussed in Shin et al. (2017), Lesort et al. (2018), and Gao & Liu (2023), leads to progressive degradation in image quality. To counter this, we propose utilizing the early-stage denoising capabilities of diffusion models to distill knowledge directly from current training data, rather than generated images. This approach yields several benefits: (1) Enhanced image clarity. (2) Preservation of knowledge from earlier stages. (3) Reduced computational cost by eliminating the need for image generation in the initial steps.

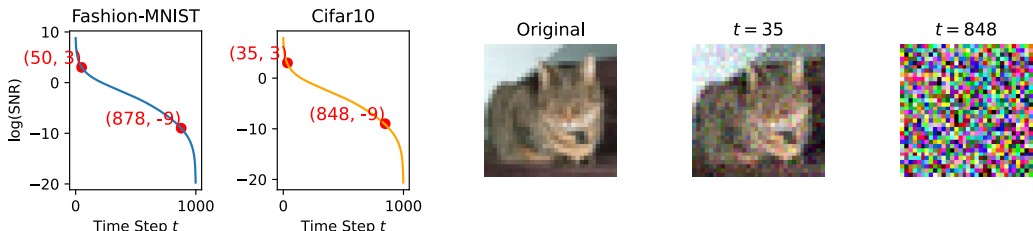

Figure 5: $logSNR$ Across Time Steps in Fashion-MNIST and CIFAR-10

To find the turning point $t_c$ of the time step before which current training data could be used, we calculate the Signal-to-Noise Ratio (SNR) along with the time step. We use the same formula as in Deja et al. (2022):

$$SNR(\boldsymbol{x}_0, t) = \frac{\bar{\alpha}_t \boldsymbol{x}_0^2}{1 - \bar{\alpha}_t} \tag{8}$$

where $\boldsymbol{x}_0$ is the original image. The SNR quantifies the amplitude ratio between the original image and noise. Research by Deja et al. (2022) demonstrates that a $log(SNR) = 3$ serves as a reliable threshold, which does not negatively impact the FID of generated images. The critical time steps, $t_{low}$, are determined as 50 for Fashion-MNIST and 35 for CIFAR-10, as shown in Fig. 5.

As the time step increases and $log(SNR)$ becomes significantly negative, indicating a strong dominance of noise over signal, the diffusion model's input approximates Gaussian noise. In such scenarios, distilling knowledge from Gaussian noise becomes crucial. We utilize a rescaled schedule, as suggested by Lin et al. (2023b), where a $log(SNR) = -9$ marks the input as nearly indistinguishable from noise. The identified transition points, $t_{high}$, are 878 for Fashion-MNIST and 848 for CIFAR-10, detailed in Fig. 5.

In the yellow region of Fig. 6, we propose selecting images for distillation based on the training step $t_r$ and two thresholds: $t_{low}$ and $t_{high}$. Specifically:

- If $t_r < t_{low}$, images are selected from the current batch for distillation.
- If $t_r > t_{high}$, images are selected from a pool generated using Gaussian noise.
- Otherwise, noisy images are generated from previous model

To manage this process, $log(SNR)$ for each image batch is calculated. Additionally, the thresholds $t_{low}$ and $t_{high}$ are dynamically updated using a moving average formula based on each training batch. This adaptive approach minimizes the need for manual tuning of these parameters and reduces the overall number of images that need to be generated by approximately 20%, without compromising performance outcomes.

## 4.5 WORKFLOW AND OVERALL OBJECTIVE

The workflow of our method is illustrated in Fig. 6, and the algorithm is shown in Algorithm 2. For a task $k$, we leverage the previously trained diffusion model $\theta^{k-1}$, the guidance classifier $g^{k-1}$, and the current dataset $\mathbb{D}_k = (\mathbf{X}^k, \mathbf{Y}^k, \mathbf{C}^k)$, which contains the images, labels, and class information, respectively.

First, we randomly select a set of time steps $t_r$ from a uniform distribution over $[0, T]$, a set of class labels $y_r$ from a uniform distribution over all encountered classes, and generate Gaussian noise $\epsilon_r$. Next, we use Sec. 4.4 to filter the selected time steps $t_r$ to determine the type of images to distill. After filtering, we prepare the images ($\mathbf{X}_r$) for distillation. We then compute the first term of the replay loss with $lwf$ representing the distillation loss proposed in Li & Hoiem (2016) for guidance classifier.

$$l_{r1} = MSE(\theta^{k-1}(\mathbf{X}_r, t_r), \theta^k(\mathbf{X}_r, t_r)) + lwf(g^{k-1}(\mathbf{X}_r, t_r), g^k(\mathbf{X}_r, t_r)) \tag{9}$$

Next, we obtain the time steps $t_{gene}$ for generating noisy images. To enhance the efficiency of the inverse process, we distill knowledge from all intermediate noisy images $X_{\tau_i}$ images, where $\tau_i$ the corresponding time step, as illustrated in the blue region of Fig. 6.

$$l_{r2} = MSE(\theta^{k-1}(\mathbf{X}_{\tau_i}, \tau_i), \theta^k(\mathbf{X}_{\tau_i}, \tau_i)) + lwf(g^{k-1}(\mathbf{X}_{\tau_i}, \tau_i), g^k(\mathbf{X}_{\tau_i}, \tau_i))] \tag{10}$$

By combining the loss computed in the first step, we obtain the overall replay loss $L_{replay}$. In our experiment, we set $\alpha = 0.2$.

$$L_{replay} = \frac{1}{1 + \alpha}(l_{r1} + \alpha l_{r2}) \tag{11}$$

Next, we sample time steps $t_c$ and noise $\epsilon_c$. We then pass the current noisy training data $(\mathbf{X}_c, \mathbf{y}_c, \epsilon_c)$ through our current model to obtain:

$$L_{current} = MSE(\theta^k(\mathbf{X}_c, t_c), \epsilon_c) + CE(g^k(\mathbf{X}_c, t_c), \mathbf{y}_c) \tag{12}$$

Finally, the overall objective is formulated as:

$$L_{total} = \frac{1}{k+1} L_{current} + (1 - \frac{1}{k+1}) L_{replay} \tag{13}$$

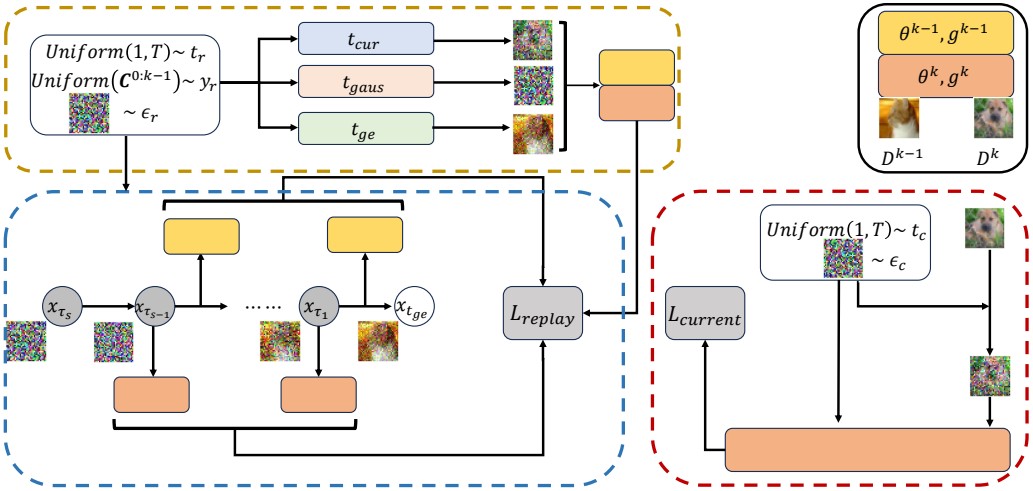

Figure 6: Illustration of Our Method. The **yellow region** represents **SGGD**, the **blue region** denotes **NIGD** and **CGGD**, and the **red region** corresponds to training on the current dataset $\mathbb{D}^k$.

# 5 EXPERIMENTS AND RESULTS

Due to page limitations, the impact of generation steps is detailed in Appendix A.4, the distribution of generated images is discussed in Appendix A.5, and the classification accuracy using generated images is examined in Appendix A.2. .

## 5.1 DATASETS

We compare our method primarily with deep generative approaches such as DGR, DGR with distillation Masip et al. (2023), and DDGR Gao & Liu (2023), along with the memory-based method ER Chaudhry et al. (2019), as well as Fine-tuning and Joint-training as lower and upper bound, respectively.

- **F.T. (Fine-tuning)**: Fine-tunes only on the current task (lower bound).
- **J.T. (Joint-training)**: Trains on all encountered tasks jointly (upper bound).

We use **DDGR-1000** with 1000 full generation steps, providing state-of-the-art performance but with high computational cost, making it a second upper bound.

For **Fashion-MNIST**, we use a small UNet Ho et al. (2020); Ronneberger et al. (2015) with 5 DDIM steps. For **CIFAR-10** and **CIFAR-100**, a medium-sized UNet with 20 DDIM steps is used. The **ER** method employs a memory buffer of size 1000.

## 5.2 EVALUATION METRICS

We assess image quality using the **Fréchet Inception Distance (FID)**, calculated between generated images and the test set of previously encountered tasks. To evaluate the model's ability to generate balanced batches, we compute the **Kullback-Leibler Divergence (KLD)** between the uniform distribution and the predicted class distribution of the generated images.

We also measure training time for all methods on a **2 × NVIDIA A100 40GB**, using **DGR-distill** as the baseline for comparison.

## 5.3 OVERALL RESULTS

In Sec. 5.3 and Fig. 7, we present our results as the mean and standard deviation over five random runs. Across all scenarios, our method outperforms DGR-distill by $3 \sim 6.1$ in FID and $0.04 \sim 0.09$ in KLD, while achieving around $15\%$ savings in computational cost.

These results demonstrate the effectiveness of our proposed comprehensive generative distillations above DGR-distill. Even compared to DDGR-1000 with 1000 generation steps, our method achieves similar performance in FID and KLD for Fashion-MNIST using only 5 steps, drastically reducing computation.

For CIFAR-10 and CIFAR-100, the performance gap between our method and DDGR-1000 slightly increases, likely due to the higher complexity of these datasets. However, using just 20 generation steps makes this gap reasonable, and increasing to 50 steps could reduce it further, as shown in Tab. 4.

Fig. 7 illustrates the evolution of FID and KLD across tasks, where our method consistently outperforms DGR-distill and approaches the performance of DDGR-1000.

# 6 ABLATION STUDY

Our method consists of three components, as detailed in Sec. 4. We begin with the baseline model, DGR-distill, and sequentially introduce each component to assess their impact, testing on Fashion-MNIST with 5 DDIM generation steps for all methods.

First, we add NIGD to the baseline, resulting in substantial improvements across both metrics. Next, adding only CGGD to the baseline primarily enhances the KLD, while NGGD mainly improves the

Table 1: Results Presented as Mean and Standard Deviation Over 5 Random Runs, with 5 Generation Steps for Fashion-MNIST and 20 for CIFAR-10 and CIFAR-100

| | Fashion-MNIST | | | CIFAR-10 | | | CIFAR-100 | | |
|---|---|---|---|---|---|---|---|---|---|
| | FID↓ | KLD↓ | Time↓ | FID↓ | KLD↓ | Time↓ | FID↓ | KLD↓ | Time↓ |
| F.T. | $65.5 \pm 8.2$ | $5.27 \pm 1.57$ | ×0.15 | $53.5 \pm 3.2$ | $1.23 \pm 0.15$ | ×0.08 | $65.6 \pm 6.7$ | $7.57 \pm 2.37$ | ×0.08 |
| DDGR-1000 | $16.2 \pm 2.1$ | $0.09 \pm 0.01$ | ×19.17 | $29.8 \pm 3.4$ | $0.13 \pm 0.02$ | ×6.85 | $34.6 \pm 4.1$ | $0.8 \pm 0.27$ | ×6.85 |
| J.T. | $14.7 \pm 1.5$ | $0.07 \pm 0.01$ | ×0.15 | $27.3 \pm 2.1$ | $0.11 \pm 0.01$ | ×0.08 | $32.4 \pm 2.5$ | $0.61 \pm 0.03$ | ×0.08 |
| ER | $20.7 \pm 0.8$ | $0.32 \pm 0.04$ | ×0.15 | $41.5 \pm 0.9$ | $0.21 \pm 0.04$ | ×0.08 | $41.6 \pm 1.4$ | $1.8 \pm 0.37$ | ×0.08 |
| DGR | $95.8 \pm 10.4$ | $1.15 \pm 0.23$ | ×0.91 | $70.3 \pm 5.2$ | $0.65 \pm 0.03$ | ×0.95 | $39.8 \pm 3.2$ | $2.6 \pm 0.57$ | ×0.95 |
| DGR-distill | $19.5 \pm 2.2$ | $0.14 \pm 0.05$ | 2.5h ×1 | $37.5 \pm 5.0$ | $0.24 \pm 0.02$ | 9.3h ×1 | $41.3 \pm 4.6$ | $1.5 \pm 0.41$ | 9.3h ×1 |
| Ours | $16.5 \pm 3.1$ | $0.10 \pm 0.04$ | ×0.85 | $32.7 \pm 3.6$ | $0.15 \pm 0.03$ | ×0.83 | $35.2 \pm 4.5$ | $0.92 \pm 0.39$ | ×0.83 |

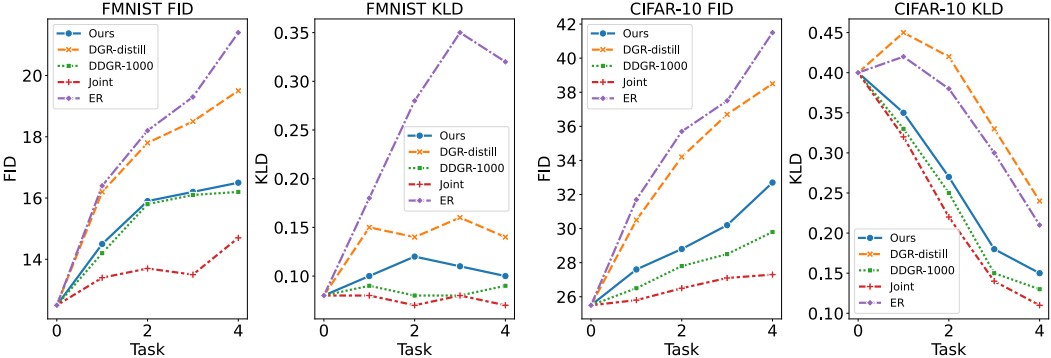

Figure 7: Evaluation of FID Score and KLD Across Tasks for Different Methods on Fashion-MNIST and CIFAR-10

FID. When all three components are combined, our method achieves significant improvements in both metrics.

Table 2: Ablation Study on Fashion-MNIST Using 5 Generation Steps

| | FID↓ | KLD↓ |
|---|---|---|
| DGR-distill | $19.5 \pm 2.2$ | $0.14 \pm 0.05$ |
| w NIGD | $17.2 \pm 2.7$ | $0.12 \pm 0.04$ |
| w CGGD | $18.4 \pm 3.2$ | $0.11 \pm 0.05$ |
| w NGGD | $18.2 \pm 1.8$ | $0.14 \pm 0.04$ |
| Ours | $16.5 \pm 3.1$ | $0.10 \pm 0.04$ |

## 7 CONCLUSION

We introduced Fast Multi-Mode Adaptive Generative Distillation (MAGD) approach, crafted to effectively mitigate catastrophic forgetting, enhance image quality, and maintain balanced class distribution in continually trained diffusion models. Incorporating Noisy Intermediate Generative Distillation (NIGD), Class-Guided Generative Distillation (CGGD), and Signal-Guided Generative Distillation (SGGD), our method not only sustains high-quality image generation across tasks but also dramatically reduces computational overhead by up to 95% for Fashion-MNIST and 88% for CIFAR, compared to traditional full-generation methods like DDGR-1000. Achieved with fewer generation steps, this performance underscores the model's efficacy in complex continual learning scenarios and its practicality for real-world applications. Future efforts will aim to expand our model's capabilities to a wider range of datasets and explore its potential in various artificial intelligence domains.

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

# A  APPENDIX

## A.1  ALGORITHM

---

**Algorithm 1** Train diffusion model at task $k$

---

**Input:** $\theta^{k-1}$, $\mathbb{D}_k$, $N_b$ is the batch size, $n$ is the number of iterations

1:  $\theta^k = \text{deepcoy}(\theta^{k-1})$
2:  **for** $n$ steps **do**
3:      sample a batch $\mathbf{X}_c, \boldsymbol{y}_c$ of size $N_b$ from $\mathbb{D}_k$
4:      $t_c, t_r \sim Uniform(\{1, \ldots, T\})$
5:      $\epsilon_c, \epsilon_r \sim \mathcal{N}(0; \boldsymbol{I})$
6:      $\mathbf{X}_c = \sqrt{\bar{\alpha}_{t_c}}\mathbf{X}_c + \sqrt{1 - \bar{\alpha}_{t_c}}\epsilon_c$
7:      $\mathbf{X}_r = \text{DDIM}(\epsilon_r, \theta^{k-1})$ {Gnerate Images from previous diffusion model}
8:      $\mathbf{X}_r = \sqrt{\bar{\alpha}_{t_r}}\mathbf{X}_r + \sqrt{1 - \bar{\alpha}_{t_r}}\epsilon_r$
9:      $l_c = \text{MSE}(\theta^k(\mathbf{X}_c, t_c), \epsilon_c)$ {current loss}
10:     **if** method == "DGR" **then**
11:         $l_r = \text{MSE}(\theta^k(\mathbf{X}_r, t_r), \epsilon_r)$
12:     **else if** method == "DGR-distill" **then**
13:         $l_r = \text{MSE}(\theta^k(\mathbf{X}_r, t_r), \theta^{k-1}(\mathbf{X}_r, t_r))$
14:     **end if**
15:     $l_t = \frac{1}{k+1}l_c + (1 - \frac{1}{k+1})l_r$
16:     $l_t.\text{backward}()$
17:     Update $\theta^k$
18:  **end for**

---

## A.2  CLASSIFICATION ACCURACY

To evaluate classification accuracy, we replace the memory buffer with our trained diffusion model, following the standard training strategies outlined in Chaudhry et al. (2019); Shin et al. (2017). For conciseness, we omit the specific implementation details from the main text.

The results are presented for three benchmark datasets: Fashion-MNIST, CIFAR-10, and CIFAR-100. For Fashion-MNIST, we use a small CNN as the classifier, while for CIFAR-10 and CIFAR-100, we employ ResNet-18.

Additionally, we introduce two new methods: **BIR**van de Ven et al. (2020), a latent distillation approach, and **PASS**Zhu et al. (2021), a memory-free method.

## A.3  DEMONSTRATION

We can reformulate Eq. (3) as follows:

$$\boldsymbol{x}_{\tau_i} = k_{\tau_{i+1}}\boldsymbol{x}_{\tau_{i+1}} + l_{\tau_{i+1}}\theta(\boldsymbol{x}_{\tau_{i+1}}) \tag{14}$$

where $k_{\tau_{i+1}} = \sqrt{\frac{\bar{\alpha}_{\tau_i}}{\bar{\alpha}_{\tau_{i+1}}}}$, and $l_{\tau_{i+1}} = \sqrt{1 - \bar{\alpha}_{\tau_i}} - k_{\tau_{i+1}}\sqrt{1 - \bar{\alpha}_{\tau_{i+1}}}$

For a DDIM process comprising $S$ steps, we have:

$$\boldsymbol{x}_{\tau_{s-1}} = k_{\tau_s}\boldsymbol{x}_{\tau_s} + l_{\tau_s}\theta(\boldsymbol{x}_{\tau_s})$$
$$\boldsymbol{x}_{\tau_{s-2}} = k_{\tau_{s-1}}\boldsymbol{x}_{\tau_{s-1}} + l_{\tau_{s-1}}\theta(\boldsymbol{x}_{\tau_{s-1}})$$
$$\ldots$$
$$\boldsymbol{x}_{\tau_i} = k_{\tau_{i+1}}\boldsymbol{x}_{\tau_{i+1}} + l_{\tau_{i+1}}\theta(\boldsymbol{x}_{\tau_{i+1}}) \tag{15}$$

---

**Algorithm 2** Train diffusion model at task $k$

---

**Input:** $\theta^{k-1}$, $g^{k-1}$, $\mathbb{D}_k$, $N_b$ is the batch size, $n$ is the number of iterations, $C$ presents the previously learned classes, $t_{low}$ is the transition point for current batch, and $t_{high}$ is the transition point for Gaussian noise.

1: $\theta = \text{deepcoy}(\theta^{k-1})$
2: $g = \text{deepcoy}(g^{k-1})$
3: **for** $n$ steps **do**
4:     sample a batch $\mathbf{X}_c, \boldsymbol{y}_c$ of size $N_b$ from $\mathbb{D}_k$
5:     $t_c, t_r \sim Uniform(\{1, \ldots, T\})$
6:     $\epsilon_c, \epsilon_r \sim \mathcal{N}(0; \boldsymbol{I})$
7:     $\hat{t}_{low} = \{t | log(SNR(\mathbf{X}_c, t)) = 3\}$
8:     $\hat{t}_{high} = \{t | log(SNR(\mathbf{X}_c, t)) = -9\}$
9:     $\mathbf{X}_c = \sqrt{\bar{\alpha}_{t_c}}\mathbf{X}_c + \sqrt{1 - \bar{\alpha}_{t_c}}\epsilon_c$ {Add noise to the current training batch}
10:     $\mathbf{X}_r, \mathbf{X}_g = [], []$ { $\mathbf{X}_r$ store the images to replay, and $\mathbf{X}_g$ represents the noisy images generated}

11:     $\boldsymbol{t}_g = []$ {time steps for the generation process}
12:     $\epsilon_{target\_gene}, \epsilon_{target\_cl} = [], []$ { taget for training diffusion model and classifier}
13:     **for** $i, t$ in enumerate($t_r$ Sec. 4.4) **do**
14:       **if** $t < t_{low}$ **then**
15:         $\mathbf{X}_r$ add $\mathbf{X}_c[i]$ {Add Current image}
16:       **else if** $t > t_{high}$ **then**
17:         $\mathbf{X}_r$ add $\epsilon_r[i]$ {Add Gaussian}
18:       **else**
19:         $y \sim Uniform(C)$
20:         $\boldsymbol{x}_g = \epsilon_r[i]$
21:         **for** j in range(S) **do**
22:           $\boldsymbol{x}_g = DDIM(\boldsymbol{x}_g, \theta^{k-1}, g^{k-1}, \tau_{s-j})$Eq. (7)
23:           $\epsilon_{target\_gene}$ ADD $\theta^{k-1}(\boldsymbol{x}_g)$
24:           $\epsilon_{target\_cl}$ ADD $g^{k-1}(\boldsymbol{x}_g)$
25:           $\mathbf{X}_g$ ADD $\boldsymbol{x}_g$
26:           $\boldsymbol{t}_g$ ADD $\tau_{s-j}$
27:         **end for**
28:       **end if**
29:     **end for**
30:     $\mathbf{X}_r = \sqrt{\bar{\alpha}_{t_r}}\mathbf{X}_r + \sqrt{1 - \bar{\alpha}_{t_r}}\epsilon_r$ {Add noise to the replay batch}
31:     $l_{current} = MSE(\theta(\mathbf{X}_c, \boldsymbol{t}_c), \epsilon_c) + CE(g(\mathbf{X}_c, \boldsymbol{t}_c), \boldsymbol{y}_c)$
32:     $l_{r1} = MSE(\theta(\mathbf{X}_r, \boldsymbol{t}_r), \theta^{k-1}(\mathbf{X}_r, \boldsymbol{t}_r)) + \text{LWF}(g(\mathbf{X}_r, t_r), g^{k-1}(\mathbf{X}_r, t_r))$
33:     $l_{r2} = MSE(\theta(\mathbf{X}_g, \boldsymbol{t}_g), \epsilon_{target\_gene}) + \text{LWF}(g(\mathbf{X}_g, t_g), \epsilon_{target\_cl})$
34:     $l_{replay} = \frac{1}{1+\alpha}(l_{r1} + \alpha l_{r2})$
35:     $l_t = \frac{1}{k+1}l_{current} + (1 - \frac{1}{k+1})l_{replay}$
36:     $l_t$.backward()
37:     Update $\theta$ and $g$
38:     $t_{low} = 0.999t_{low} + 0.001\hat{t}_{low}$
39:     $t_{high} = 0.999t_{high} + 0.001\hat{t}_{high}$
40: **end for**
41: **return** $\theta$ and $g$

---

Starting from the initial step, with $\tau_s = 999$ for a total of 1000 steps, $\boldsymbol{x}_{\tau_s}$ represents random noise. Based on the recurrence relation, we obtain:

$$\boldsymbol{x}_{\tau_i} = \sqrt{\frac{\bar{\alpha}_{\tau_i}}{\bar{\alpha}_{\tau_s}}}\epsilon + \sqrt{\frac{\bar{\alpha}_{\tau_i}}{\bar{\alpha}_{\tau_{s-1}}}}l_{\tau_s}\theta(\boldsymbol{x}_{\tau_s}) + \cdots + \sqrt{\frac{\bar{\alpha}_{\tau_i}}{\bar{\alpha}_{\tau_{i+1}}}}l_{\tau_{i+2}}\theta(\boldsymbol{x}_{\tau_{i+2}}) + l_{\tau_{i+1}}\theta(\boldsymbol{x}_{\tau_{i+1}}) \quad (16)$$

Table 3: Results Presented as Mean and Standard Deviation Over 5 Random Runs, with 5 Generation Steps for Fashion-MNIST and 20 for CIFAR-10 and CIFAR-100

|  | Fashion-MNIST | CIFAR-10 | CIFAR-100 |
|---|---|---|---|
|  | Acc↑ | Acc↑ | ACC↑ |
| F.T. | $17.3 \pm 2.1$ | $19.5 \pm 0.1$ | $16.5 \pm 2.2$ |
| DDGR-1000 | $83.2 \pm 1.1$ | $44.3 \pm 1.2$ | $34.5 \pm 0.7$ |
| i.i.d. Off | $92.3 \pm 0.3$ | $83.2 \pm 0.1$ | $67.4 \pm 0.3$ |
| ER | $79.4 \pm 2.4$ | $28.3 \pm 2.4$ | $25.1 \pm 1.2$ |
| DGR | $57.4 \pm 5.3$ | $27.5 \pm 3.5$ | $23.5 \pm 2.4$ |
| DGR-distill | $75.2 \pm 3.1$ | $35.4 \pm 4.1$ | $28.7 \pm 1.1$ |
| BIR | $78.5 \pm 2.8$ | $36.1 \pm 5.7$ | $21.7 \pm 0.4$ |
| PASS | $79.7 \pm 3.7$ | $39.2 \pm 3.2$ | $30.3 \pm 0.8$ |
| Ours | $80.4 \pm 4.1$ | $41.5 \pm 2.8$ | $32.1 \pm 1.2$ |

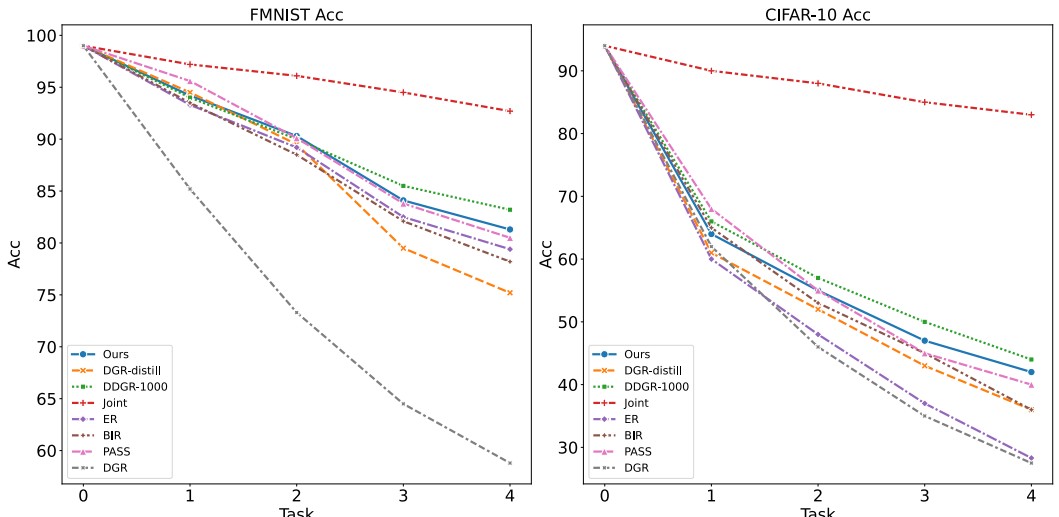

Figure 8: Evaluation of the Final Classification Accuracy Across Tasks for Different Methods on Fashion-MNIST and CIFAR-10

$$\boldsymbol{x}_0 = \sqrt{\frac{\bar{\alpha}_0}{\bar{\alpha}_{\tau_s}}}\epsilon + \sqrt{\frac{\bar{\alpha}_0}{\bar{\alpha}_{\tau_{s-1}}}}l_{\tau_s}\theta(\boldsymbol{x}_{\tau_s}) + \cdots + \sqrt{\frac{\bar{\alpha}_0}{\bar{\alpha}_{\tau_1}}}l_{\tau_2}\theta(\boldsymbol{x}_{\tau_2}) + l_{\tau_1}\theta(\boldsymbol{x}_{\tau_1}) \tag{17}$$

By introduce Eq. (17) into Eq. (4) and minus Eq. (16), we derive:

$$\hat{\boldsymbol{x}}_{\tau_i} = \boldsymbol{x}_{\tau_i} + \sum_{j=i}^{1}(\boldsymbol{r}_j\theta(\boldsymbol{x}_{\tau_j})) \tag{18}$$

where:

$$\boldsymbol{r}_j = \sqrt{\frac{\bar{\alpha}_{\tau_i}}{\bar{\alpha}_{\tau_{j-1}}}}l_{\tau_j} = \sqrt{\bar{\alpha}_{\tau_i}}(\sqrt{\frac{1-\bar{\alpha}_{\tau_{j-1}}}{\bar{\alpha}_{\tau_{j-1}}}} - \sqrt{\frac{1-\bar{\alpha}_{\tau_j}}{\bar{\alpha}_{\tau_j}}}) \tag{19}$$

## A.4 The Influence of Generation Steps

In this section, we analyze the impact of varying generation steps on our method. As shown in Tab. 4, our method consistently outperforms the baseline (DGR-distill) by a large margin in both

FID and KLD across all generation steps. Notably, our method with only 20 steps achieves FID scores close to those of DGR-distill with 100 steps. Furthermore, our method with 50 steps closely matches DDGR-1000, achieving FID scores of 30.8 **vs** 29.8 and KLD of 0.13 **vs** 0.13.

Table 4: FID and KLD on CIFAR-10 Across Different Generation Steps

| Steps | 5 | | 10 | | 20 | | 50 | | 100 | |
|---|---|---|---|---|---|---|---|---|---|---|
| | FID↓ | KLD↑ | FID↓ | KLD↑ | FID↓ | KLD↑ | FID↓ | KLD↑ | FID↓ | KLD↑ |
| DGR-distill | 45.4 ± 3.8 | 0.47 ± 0.05 | 41.3±4.4 | 0.33±0.05 | 37.5±5.0 | 0.24±0.02 | 35.1 ±2.8 | 0.18 ±0.02 | 31.5±1.9 | 0.15 ±0.03 |
| Ours | 40.5± 4.7 | 0.28 ± 0.06 | 35.8± 4.1 | 0.20 ±0.05 | 32.7 ± 3.6 | 0.15 ±0.03 | 30.8 ± 4.5 | 0.14 ±0.02 | 30.1±2.5 | 0.13 ± 0.03 |

## A.5 THE DISTRIBUTION OF GENERATED IMAGES

We analyze the distribution of generated images by our method and DGR-distill on Fashion-MNIST, as shown in Fig. 9. This analysis is based on one experimental run. Task 0 includes classes $[7, 9]$, and the left figure of Fig. 9 illustrates the image distribution after the first task, where both methods operate identically. We observed fewer images of class 7 during this task.

The middle figure shows the proportion of class 7 in the generated images after training on each task. With DGR-distill, the proportion rapidly decreases and nearly disappears by the final task due to error accumulation from replaying only generated images. In contrast, our method uses a simple guided-classifier to maintain balanced image generation, keeping class 7 at a stable proportion.

The right figure compares the overall KLD, where our method significantly outperforms DGR-distill, generating more balanced images across all classes.

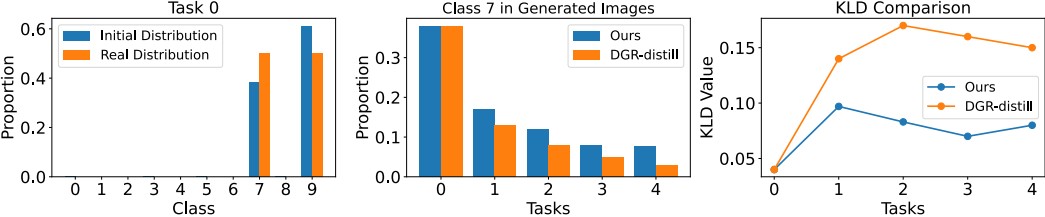

Figure 9: Comparative Evaluation of Image Distribution Generated by Our Method and DGR-Distill on Fashion-MNIST

