# OpenReview forum: "Fast Multi-Mode Adaptive Generative Distillation for Continually Learning Diffusion Models"
_ICLR.cc/2025/Conference — ICLR 2025 Conference Withdrawn Submission_

### Official Review · Reviewer_hQXi · 2024-11-01

**Soundness:** 2
**Presentation:** 2
**Contribution:** 2
**Rating:** 3
**Confidence:** 4

**Summary:**

The paper solves the catastrophic forgetting problem for the diffusion model in a continual learning fashion on images with class labels. The paper proposes several techniques including Noisy Intermediate Generative Distillation (NIGD), Class-guided Generative Distillation (CGGD), and Signal-Guided Generative Distillation (SGGD). The method shows ablative improvements on small-scale datasets like CIFAR and Fashion-MNIST.

**Strengths:**

* The paper provides enough details in the methodology.
* The experiments provide several ablation studies for the proposed components.

**Weaknesses:**

* The problem setup seems to have a slightly narrow scope, which may be less attractive to the ICLR audience.
* The motivation for solving such a problem is not well-justified in the introduction. I am not very convinced if this is a realistic setup or an ad-hoc problem.
* The datasets in the experiments are quite small and the example of continual learning is synthetic and somewhat toy.
* The baselines look outdated and not very comprehensive.
* The writing does not seem very mature due to issues like jumping logic, inconsistent notations, and wrong citing formats. More proofreading and revision might be needed.

**Questions:**

Can the method work on larger-scale datasets?

---

### Official Review · Reviewer_h6Pg · 2024-11-04

**Soundness:** 2
**Presentation:** 2
**Contribution:** 2
**Rating:** 3
**Confidence:** 3

**Summary:**

The paper introduces Noisy Intermediate Generative Distillation (NIGD), Class-Guided Generative Distillation (CGGD), and Signal-Guided Generative Distillation (SGGD), which not only significantly improve quantitative results but also reduce the computational cost of Diffusion Models in Continual Learning settings.

**Strengths:**

SGGD is a good strategy to choose source of input images based on analysis from Signal-to-Noise Ration (SNR).

**Weaknesses:**

The article doesn't seem well written and organized.

1. In the NIGD section (4.2), Figure 4 shows that the weight of the residual term is less than 1 at small steps ($i < 10$ for 20-step DDIM) but significantly increases at larger steps. The authors also note that they use all intermediate noisy images generated throughout the inverse process (line 299). Could this lead to a significant difference between noisy images and directly generated images? Previous work [1, 2] indicates a gap between the forward and backward processes of Diffusion Models. Since NIGD is proposed to reduce computation costs, a more detailed analysis is necessary.

2. CGGD (4.3) appears to be directly incorporated from previous works [3, 4].

3. (Minor) Typo: 'Generate Images' in line 7 of Algorithm 1.

[1] On Inference Stability for Diffusion Models

[2] Refining Generative Process with Discriminator Guidance in Score-based Diffusion Models

[3] Diffusion Models Beat GANs on Image Synthesis

[4] DDGR: Continual Learning with Deep Diffusion-Based Generative Replay

**Questions:**

1. Could the author provide more comprehensive experimental results, for example, on the ImageNet and text-to-image datasets, at least ImageNet $64 \times 64$ like DDGR[1]?

2. Classification accuracy is presented in the Appendix. Can the author provide additional metric used in Continual Learning, such as forgetting rate?

3. Are there any insights of choice for weight of each components in $L_{replay}$ (Equation 11) and $L_{total}$ (Equation 13)?

[1] DDGR: Continual learning with deep diffusion-based generative replay

---

### Official Review · Reviewer_kSJy · 2024-11-06

**Soundness:** 3
**Presentation:** 2
**Contribution:** 2
**Rating:** 5
**Confidence:** 3

**Summary:**

This paper addresses the setting of using diffusion models as generative replay in continual learning, and proposes a framework, MAGD, for continually updating the diffusion models effectively. It proposes three distillation strategies: distilling from intermediate noise images, using classifier guidance, and using different distillation image types based on sampled time steps. Results show that MAGD achieves strong performance with fast computation time.

**Strengths:**

* This paper tackles an interesting and important topic of continual learning for diffusion models.
* The proposed framework is reasonable from a technical standpoint and the explanations are detailed.
* The evaluation encompasses a variety of baselines and datasets, and provides fine-grained analysis by tasks.

**Weaknesses:**

It would be helpful to have more clarity and analysis on the benefits and necessity of the proposed distillation approaches. Specifically:
* In the paper, various experimental results seem to suggest that the performance may be influenced more by the quality of diffusion sampling than by specific choices of distillation strategies. For instance, while DDGR-1000 achieves superior results, its core approach is comparable to DGR with classifier guidance. Table 4 also shows that when a higher number of diffusion steps (100) is used, the performance gap between DGR-distill and MAGD becomes significantly smaller. These could imply that when sample quality is sufficiently high, the impact of different distillation strategies on performance becomes small.
* On the speed aspect, there are fast samplers like [1,2] that can be applied off-the-shelf to generate high quality samples in few steps. This raises the question of whether existing methods like DDGR are truly slow, or if they could be easily sped up with minimal impact on performance. It could be helpful to see comparisons under using faster samplers.

Additional minor points:
* It could be helpful to provide qualitative visualizations and comparisons of the diffusion generated images, to facilitate clarity and understanding.
* The presentation feels a bit rough around the edges - there are a number of typos and missing spaces or punctuations (L210, L255, L303, L330, etc.); supplementary Table 3 should be centered.

[1] Lu et al. "DPM-Solver: A Fast ODE Solver for Diffusion Probabilistic Model Sampling in Around 10 Steps." (NeurIPS 2022).

[2] Lu et al. "Dpm-solver++: Fast solver for guided sampling of diffusion probabilistic models." (arXiv 2022).

**Questions:**

Please see weaknesses above.

---

### Official Review · Reviewer_hJjz · 2024-11-07

**Soundness:** 3
**Presentation:** 3
**Contribution:** 2
**Rating:** 5
**Confidence:** 3

**Summary:**

The proposed method aims to solve the computation-demanding issue of diffusion models in CL, where relatively rapid adaptation is crucial. The authors look into the diffusion process and propose altering different aspects for better image quality and training speed by proposing three major components: NIGD, CGGD, and SGGD.

**Strengths:**

1. The proposed method dives deep into the diffusion process and improves diffusion model training in CL scenarios. Specifically, the SGGD is interesting.
2. The proposed method outperforms both diffusion-based methods and ER-based methods.
3. The proposed method improves the training time compared to previous arts.
4. Decent writing and figures that illustrate the ideas and intuition of the paper.

**Weaknesses:**

1. despite aiming to adapt diffusion models for rapid update during continual learning, the proposed method still requires several hours of training time.

2. The authors only compare to plain ER, and the accuracy is within one standard deviation range. This questions the applicability of the proposed method for image classification in CL, given the drastically more training time and memory required.

3. The proposed CGGD does not seem novel. CGGD simply leverages classifier guidance for class balancing.

4. Notation
(1) In Eq. 1: Should the summation of losses be N instead of n?
(2) Line 232, What is the output? Is it the noised image or the noise itself? This needs to be clarified. (Based on Algo. 1, it seems to be noise?)

**Questions:**

1. I do not completely understand why leveraging the direct approach could mitigate the degradation of the diffusion model in CL. If the two are equivalent for distillation (as mentioned by the authors in Line 291), then why are there additional benefits and higher-quality images? Elaboration from the authors could be helpful.

2. I think a qualitative demonstration is missing here. Specifically, how are the images generated by the proposed method in different stages? Because it seems the main target of the work is for generation and not for improving the classification accuracy (while in Table 3, there is a slight improvement in accuracy compared to a simple baseline like ER, I doubt the performance would surpass other SoTA replay-based methods). Hence, the demonstration of the generated images seems important under this scenario. The drastically more computation time required is also a disadvantage if one were to use a diffusion model as a replay.

3. While the proposed method shows improvement in KL Divergence and FID over the baselines on three benchmarks, these benchmarks are commonly deemed unrealistic (small size and resolution). I wonder how the proposed method performs in more realistic datasets, e.g., ImageNet.

---

### Note · Authors · 2024-11-14

I have read and agree with the venue's withdrawal policy on behalf of myself and my co-authors.